# Antioxidative Properties of Melanins and Ommochromes from Black Soldier Fly *Hermetia illucens*

**DOI:** 10.3390/biom9090408

**Published:** 2019-08-23

**Authors:** Nina Ushakova, Alexander Dontsov, Natalia Sakina, Alexander Bastrakov, Mikhail Ostrovsky

**Affiliations:** 1A.N. Severtsov Institute of Ecology and Evolution of Russian Academy of Sciences, 119071 Moscow, Russia; 2N.M. Emanuel Institute of Biochemical Physics of Russian Academy of Sciences, 119334 Moscow, Russia

**Keywords:** black soldier fly, melanins, ommochromes, antioxidant activity, ESR

## Abstract

A comparative study of melanin and ommochrome-containing samples, isolated from the black soldier fly (BSF) by enzymatic hydrolysis, alkaline and acid alcohol extraction or by acid hydrolysis, was carried out. Melanin was isolated both as a melanin-chitin complex and as a water-soluble melanin. Acid hydrolysis followed by delipidization yielded a more concentrated melanin sample, the electron spin resonance (ESR) signal of which was 2.6 × 10^18^ spin/g. The ommochromes were extracted from the BSF eyes with acid methanol. The antiradical activity of BSF melanins and ommochromes was determined by the method of quenching of luminol chemiluminescence. It has been shown that delipidization of water-soluble melanin increases its antioxidant properties. A comparison of the antioxidant activity of BSF melanins and ommochromes in relation to photoinduced lipid peroxidation was carried out. The ESR characteristics of native and oxidized melanins and ommochromes were studied. It is assumed that *H. illucens* adult flies can be a useful source of natural pigments with antioxidant properties.

## 1. Introduction

The study of biologically active substances of insects is important in connection with the search for new types of drugs and pharmaceutical raw materials. To the number of biologically active biomolecules synthesized by insects can be included the nitrogen-containing pigments melanins and ommochromes. They are part of the cuticle, determining its color, and are present in the hemolymph, in the structures of the complex eye, in the intestine and in the fat body [1]. The pigments play the role of cellular photo- and radioprotectors, antioxidants, endogenous regulators of redox reactions that ensure the ability of organisms to survive in the external environment, including under extreme conditions [2,3,4,5,6]. Melanins plays an important role in the human and animal body, protecting from the damaging effects of extreme environmental factors. The inhibitory effect of melanins on the process of lipid peroxidation caused by various prooxidants is well known [2,7,8,9,10]. The mechanism of the protective action of melanins is associated with their ability to scavenge free radicals, inactivate ROS and bind prooxidant ions metals of transition valence into inactive complexes [2,11,12]. Ommochromes are phenoxazine pigments. They are divided into two main subgroups—ommatins and ommins—which are dimers and oligomers of kynurenine derivatives, respectively [13,14]. Ommochromes strongly absorb ultraviolet, and also have characteristic absorption bands in the visible range at ~430–530 nm, and their absorption maxima depend on the redox state and the pH value of the medium. In vivo, both ommatins and ommins appear dark in color; they may be yellow brown or purple. Invertebrate animals use ommochrome pigments to color various parts of the body and for mimicry. It was shown that xanthommatin and its derivatives, such as, for example, decarboxylated xanthommatin and ommatin D, are responsible for the color and its changes in arthropods [15,16,17]. Ommochromes are characteristic pigments of the eyes of insects and other arthropods. It is believed that the main functions of ommochromes are screening the photoreceptors from diffused light, antioxidant protection of retinular and pigment cells, as well as cutaneous covering staining and detoxification of excess tryptophan [6,18,19]. The antioxidant activity of ommochromes was previously shown experimentally for pigments from the eyes of crustaceans [8,20], some insect species [8,21], and also theoretically for ommatin D [22] and their precursor kynurenin [23].

A characteristic feature of melanins is the presence of unpaired electrons in these molecules, which determines their paramagnetic properties [24]. Therefore, the electron spin resonance method (ESR) is an important tool for identifying and studying the free radical properties of these pigments. The ESR method allows one to identify the chemical nature of free radicals by the magnitude of the effective spectroscopic splitting factor (g-factor), which is different for various classes of free radicals [25]. In addition, the ESR method allows one to estimate the number of free radicals in a sample by the intensity of the spectral signal, which is directly proportional to the concentration of radicals, and their lifetime by the width of the spectral line (∆H), which contains information on the relaxation times of excited spin states [25].

There is currently increased attention to edible insects as they can be a source of protein, fats and biologically active compounds. There is a great interest in black soldier fly *Hermetia illucens*, which can develop in a pure culture under artificially controlled conditions. This insect is promising for the industrial production of biomass, the processed products of which can be used in feed, food and also in medicine and cosmetology [26,27,28]. Although melanins and ommochromes are relatively well studied, the biochemical features of the black soldier fly require a more detailed study of the properties of the pigment and its isolation methods.

The purpose of this work is isolation and study the properties of melanins and ommochromes obtained from adults of black soldier fly *H. illucens*.

## 2. Materials and Methods

### 2.1. Isolation of Melanin From BSF

The object of the study was dead adult flies of *H. illucens* after breeding under laboratory conditions. Pure BSF culture is maintained in A.N. Severtsov Institute of Ecology and Evolution, Moscow. The technology of insect cultivation consists of several stages: keeping adult flies in insectarium under controlled conditions, incubating eggs and obtaining larvae in an incubator, growing larvae in containers with a nutrient substrate, and obtaining prepupae, then pupae from which adult flies fly. In insectarium adult flies are kept at a temperature of 28–35 °C and humidity of 60%; daylight lasts 12 h when illuminated with a fluorescent lamp with an intensity of at least 5000 lux. Adult flies live 5–8 days. The nucleotide sequence of *Hermetia illucens* is registered at GenBank (ncbi.nlm.nih.gov/genbank/), (*Hermetia illucens*, sample H-il 1) No. KY817115. Dead flies were collected after the end of their life cycle and stored frozen at −18 °C.

In our experiments, the following scheme was used to process the biomass of flies:


*a. preparation of water-soluble melanin from BSF (WSM preparation)*
-homogenization of thawed biomass in a mixer (laboratory disperser HG-202, Vilitek, Moscow, Russia);-extraction of water-soluble melanin using 10% NaOH in a weight/volume ratio of 1:4 at room temperature for 16 h;-filtering the alkaline extract through a paper filter (ash less filter paper White Ribbon, Moscow, Russia);-precipitation of water-soluble melanin from the extract by acidifying it to pH 2.0 by concentrated hydrochloric acid. The precipitate was washed with distilled water to neutral pH, followed by centrifugation at least 3 times, and then dried at 70 °C to constant weight (water-soluble melanin WSM).



*b. preparation of chitin-melanin complex from BSF (AHM preparation)*
-homogenization of thawed biomass in a mixer;-acid hydrolysis of biomass with sulfuric acid. For this, 20 g of homogenized biomass of the flies was mixed with 150 mL of 25% sulfuric acid and boiled at reflux for 3 h;-the hydrolysate was cooled to 40 °C and filtered on a Buchner funnel with a water-jet pump, then washed with distilled water. After drying in a desiccator (Medsteclo, Klin, Russia), a black, oily to the touch powder was obtained. The mass of this powder was 5.8 g, which corresponds to a yield approximately 28% (AHM preparation).


To remove fat, 10 mL of chloroform was added to WSM and AHM melanin samples (400 mg each) and incubated with shaking for 1 h. After settling the mixture, the chloroform extract was carefully removed with a glass pipette, the chloroform residue was removed with evaporation in a boiling-water bath (60–70 °C), and the delipidated samples were air dried. Dried samples were ground up in a mortar and used in EPR measurements and to determine antioxidant activity. The delipidized melanin samples are marked as DL-WSM and DL-AHM.

Synthetic DOPA-melanin was obtained by the oxidative polymerization of dioxyphenylalanine in a weakly alkaline medium (pH 10), followed by purification by the standard method [29].

### 2.2. Isolation of BSF Ommochromes

To isolate ommochromes from dead adult flies, the heads were separated manually and kept frozen at −18 °C. The thawed heads of the flies were pre-extracted with absolute methanol to separate pigments soluble in neutral methyl alcohol. For this, 300 mL of absolute methanol was added to 10.2 g of heads of the flies and incubated at room temperature and shaken periodically for 24 h. The methanol extract was discarded, and 500 mL absolute methyl alcohol containing 1% by volume of dry HCl (MeOH-HCl) was added to the remaining mass of BSF heads. The mixture was thoroughly shaken and left for extraction for 48 at a temperature of 6 °C. After that, the extract was filtered through a filter paper (Whatman qualitative filter paper, Grade 6, Sigma-Aldrich, St Louis, USA). The resulting supernatant intensely burgundy color was neutralized with 20% NH_4_OH to neutral pH. The resulting loose precipitate was separated by centrifugation at 5000× *g* for 15 min and redissolved in 100 mL of fresh MeOH-HCl solution. This procedure was repeated twice. The obtained precipitate of ommochromes was washed with distilled water and dried in a desiccator over anhydrous calcium chloride. As a result, 400 mg of a dry preparation was extracted. For the experiments, both molecular solutions of ommochromes in MeOH-HCl and finely dispersed suspensions of ommochromes in 0.1 M potassium phosphate buffer, pH 7.4, were used. Optical absorption spectra were measured on a spectrophotometer UV—1601PC, Shimadzu, Japan.

### 2.3. The Measurements of Antiradical and Antioxidant Activities

The antiradical activity of the isolated pigments of melanins and ommochromes was determined by a homogeneous hydrophilic chemiluminescent system consisting of hemoglobin, hydrogen peroxide and luminol [30]. As the measured parameters, the latent period of reaching the maximum chemiluminescence intensity was taken. Chemiluminescence kinetics were recorded on a spectrofluorometer RF 5301PC, Shimadzu, Japan) at a luminescence wavelength of 470 nm at room temperature. To quantify the ability of pigments to interact with radicals localized in the aqueous phase of this model system, chemiluminescence quenching results were recalculated in the coordinates of the dependence of latent period on the concentration of the pigment and compared with the same dependence for ascorbate, for which constant of chemiluminescence quenching is known under these conditions. The amount of anti-radical activity of the pigment was expressed as the molar concentration of ascorbate, causing the same inhibitory effect as 1 mg/mL of pigment. The incubation medium contained 0.05 M K-phosphate buffer, pH 7.4, 2 μM hemoglobin, 100 μM luminol, 100 μM EDTA (ethylene diamine tetraacetate), and various concentrations of melanins or ommochromes in K-phosphate buffer, pH 7.4 or in a solution of methanol-HCl. The reaction was started by adding 100 μM hydrogen peroxide. Control samples were samples containing a buffer solution without pigments.

The antioxidant activity of melanins and ommochromes was assessed by their inhibitory effect on the peroxidation process of the photoreceptor outer segments (POS) eye cells of the bull. Peroxide oxidation of POS was induced by ascorbate in the presence of exogenous or endogenous ferrous iron ions. The development of the process was judged by the accumulation of reaction products that reacted with thiobarbituric acid (TBA)—TBA-reactive products (TBARS), the concentration of which was determined by the standard method according to [31]. The reaction medium contained 2.5 mL of 0.1 M K-phosphate buffer, pH 7.4, 40 μg/mL of rhodopsin POS bovine, 0.5 mM ascorbate, and 0.4 mg/mL of fly melanin or 0.35 mg/mL of fly ommochromes in phosphate buffer. The reaction time was 5, 10 and 15 min.

### 2.4. The Measurements of Sorption Capacity of Melanins

The sorption capacity of the melanins of BSF was determined in relation to the binding of the methylene-blue dye and it was compared with the sorption capacity of the synthetic DOPA-melanin. Incubation of melanins with dye solutions was made at room temperature for one hour, after which aliquots of the samples were centrifuged and the concentration of free methylene-blue was measured spectrophotometrically at 665 nm. The dependence of melanin-bound methylene-blue on the value of its reverse concentration was plotted. The value of the sorption capacity of melanin was expressed in mg of bound dye by one gram of pigment when approximating the concentration of the dye to infinity.

### 2.5. Oxidative Destruction of Pigments

It is known that melanins under the action of strong oxidizing agents (hydrogen peroxide, potassium peroxide) become discolored and lose their antioxidant properties [32] simultaneously with a sharp decrease in the intensity of the ESR signal [33]. Hydrogen peroxide also causes the destruction of the ommochromes of the eye of Drosophilae [34]. Oxidative destruction of melanins and ommochromes from flies was caused by 1.0–1.5% hydrogen peroxide. Suspension of pigments (2–3 mg/mL) in 0.1 M K-phosphate buffer (pH 7.4) were then incubated in the presence of hydrogen peroxide for 2 h. After that, we compared the absorption and ESR spectral characteristics of the original and oxidized pigments.

### 2.6. ESR Spectroscopy

Because melanin is a paramagnetic polymer containing a population of stable organic free radicals, ESR spectroscopy is a particularly effective method for studying them [35,36]. There is probably only one publication that showed paramagnetism of ommochromes isolated from the eyes of crustaceans [37]. Measurements of the ESR spectra were made on the spectrometer Bruker EMX, Bruker, Germany. The spectra were recorded for both dry samples, weighing 50 mg that were placed in a cylindrical quartz cuvette at room temperature and for frozen pigment suspensions at liquid nitrogen temperature. The conditions for recording the ESR spectra were as follows: the modulation amplitude was 1.25–3.0 Gs, the sweep range was 50 Gs, the microwave frequency was 9.8 GHz, the microwave power is 0.2 mW, and the time constant was 100 ms.

### 2.7. Statistical Analyses

The data were expressed as the mean ± SD. For the statistics, Student’s t-test was used. *p* < 0.05 was considered as statistically significant.

## 3. Results

The *H. illucens* fly contains melanin, both in water-soluble form and in the form of a water-insoluble complex with chitin [38]. In this case, all the investigated samples contain lipids. Samples of water-soluble melanin (WSM) in both the initial and delipidated form (DL-WSM) were well solubilized in phosphate buffer and even in water. All melanin samples obtained (Table 1) had ESR spectra characteristic of this pigment, with small variations [39]. The highest concentration of paramagnetic centers, reaching about half of that for synthetic DOPA-melanin, was observed for the delipidated form of the melanin-chitin sample obtained by acid hydrolysis (DL-AHM). From Table 1 it also follows that treatment with chloroform leads to an increase in the concentration of paramagnetic centers in the delipidated samples. The characteristic ESR spectrum of the water-soluble melanin of the BSF is shown in Figure 1. Interestingly, the oxidation of this melanin with hydrogen peroxide apparently leads to the breakdown of the pigment, since the ESR signal almost completely disappears (Figure 1, curve 2). The results of the study of the antiradical and sorption properties of melanins are presented in Table 2. It can be seen that water-soluble melanin shows the maximum antiradical and sorption activity, especially in its delipidated form (DL-WSM). Moreover, its sorption activity is even higher than that of synthetic DOPA-melanin (Table 2).

BSF ommochromes, apparently, belong to the class of ommatins. The absorption spectrum of purified ommochromes has an absorption maximum (shoulder) in the visible region approximately under 470 nm. This absorption maximum is characteristic of dihydroxanthommatin, which is widespread in insects (Figure 2, curve 1). Oxidation of this sample with hydrogen peroxide leads to the disappearance of the absorption maximum at 470 nm, as is the case with the oxidation of dihydroxanthommatin to xanthommatin and, subsequently, probably, to the destruction of the pigment (Figure 2, curves 2 and 3), which can be seen from a sharp decrease in the concentration of paramagnetic centers in the ESR spectrum (Figure 3, curve 2). Therefore, we can assume that one of the pigments of ommochrome’s fraction of BSF is xanthommatin. It is not excluded, however, that other pigments from the class of ommatins and even ommins are also included in the fraction of ommochromes of the BSF. The BSF ommochromes show a pronounced ESR signal, the parameters of which are presented in Table 3. It can be seen that the concentration of paramagnetic centers is comparable to the concentration of the preparation melanin-chitosan complex which was obtained by acid hydrolysis (AHM), but have twice a large signal’s half-width. The ESR spectrum of BSF ommochromes practically did not differ from the ESR spectrum of BSF melanins (Figure 3), and the oxidation of ommochromes with hydrogen peroxide also leads to a significant decrease in the concentration of paramagnetic centers.

Isolated fly pigments have good antioxidant and antiradical activity. Figure 4 shows that both ommochromes (curve 1) and water-soluble melanin (curve 2) increase the latent period of development of chemiluminescence of luminol depending on the concentration. Moreover, the concentration of pigments exhibiting high inhibitory activity is in the microgram range. Interestingly, despite lower antiradical activity, the sample of water-soluble melanin was more effective in suppressing ascorbate-induced peroxidation of the outer segments of photoreceptor cells. Figure 5 shows that both water-soluble melanin and ommochromes effectively inhibit the process of lipid peroxidation. The reason for the more antioxidant activity of water-soluble melanin compared to ommochromes is not clear. Perhaps it is associated with the high sorption activity of melanin and effective binding of endogenous prooxidant metal ions of transition valence [11,12].

## 4. Discussion

In fly *H. illucens*, melanin is present in at least two different types: in the form of a water-insoluble complex with chitin and in the form of a chitin-free water-soluble pigment. The ESR characteristics of both types of BSF melanin samples are typical of eumelanins. So, the g-factor BSF melanins is in the range of 2.0037–2.0040 (Table 1), which indicates that the paramagnetic properties of these melanins may be of a semiquinone nature [40]. This is fully consistent with the hypothesis that the paramagnetic properties of eumelanins are also due to the contribution of semiquinone free radicals, which are formed as a result of the proportional distribution between hydroquinone and indole quinone in the structure of the melanin molecule [41].

Melanins—both water-soluble (WSM preparation) and chitin-melanin complex (AHM preparation) when isolated from BSF—contain impurity lipids. It was previously shown that lauric acid dominates in the lipid composition of the melanin–lipid complex of pupae [42]. The presence of lipids reduces the antioxidant properties of water-soluble melanin, which requires special defatting operations. The selection of melanin by acid hydrolysis followed by delipidization allows one to obtain more concentrated melanin (preparation DL-AHM), the ESR signal is 26 × 10^17^ spin/g. However, their antioxidant activity is relatively low that it is probably connected with its poor solubility. In addition, the low value of the antioxidant activity of the melanin sample after acid hydrolysis may be associated with the presence of a large amount of calcium contained in this preparation, which leads to blockading of binding sites for positive metal ions and, as a result, less sorption activity of the AHM preparation. It is known that the cuticle of BSF is hard and rich in calcium salts [43], which suggests the future development of a technique for removing calcium from BSF biomass. The results show that water-soluble melanin samples, although they contain a relatively small amount of melanin, can be used as antioxidants and biosorbents. Antioxidants are used as food additives and for the introduction in food products in order to prevent their deterioration, increasing shelf life.

Extraction of BSF heads with absolute methanol containing 1% dry HCL leads to the isolation of pigments exhibiting properties typical of ommochromes [34,44]. BSF has proven to be an excellent natural source of ommochromes. According to our data, they contain about 3% of ommohromes from the dry weight of the heads of flies. Here, for the first time, we cite evidence that BSF ommochromes contain a high concentration of stable free radical centers, which allows us to treat ommochromes like melanins as scavengers of active free radicals (Table 3, Figure 3). The g-factor of BSF ommochromes (2.0045, Table 3) is in the range 2.0040–2.0050, which is characteristic of phenoxy radicals [40]. It is known that the intermediates of phenoxazine, which is part of the structure of the ommochrome molecules, produce a stable ESR signal [45,46]. Therefore, it is similarly very likely that the stable ESR signal of BSF ommochromes that we discovered is due to the presence of exactly precisely the phenoxazine ring in their structure. Moreover, it is possible that it is phenoxazine that determines the antioxidantive and antiradical activity of ommochromes [47].

The BSF antioxidant mechanism of ommochromes can be due both to their reaction with active forms of oxygen, and to their ability to utilize free radicals [6,8,22,23]. BSF ommochromes can also be used as natural antioxidants and as food coloring, which additionally have antioxidant properties. A new perspective that requires additional study is the antimicrobial effect of BSF ommochromes (in press). In addition, the study of melanins and ommochromes of invertebrates and vertebrates is important for finding out their biological functions as light filtering and antioxidant pigments.

Global current challenges include a deficiency in nutritious dietary protein and the accumulation of organic waste. The most important task is the search for new sources of protein for human nutrition and animal feed, and the disposal of food waste is the most important environmental problem. It is believed that by 2050 the protein of industrially cultivated insects can reach 15% of the total amount of protein produced in the world. The use of *H. illucens* flies is promising for both of these directions, since its larvae, on the one hand, are a source of high-grade protein, and on the other hand, they can utilize food waste throughout their life [48,49]. However, when breeding these insects on an industrial scale, secondary waste is formed and accumulated in the form of biomass of dead flies. This biomass can be an essential renewable source of pigments—melanins and ommochromes.

Thus, if there is a large production of BSF breeding in order to obtain food protein, the biomass of dead adult flies will accumulate. A relatively simple method of preparing pigments made it possible to pick out natural pigments melanins and ommochromes from the industrial *H. illucens* flies, which can be successfully used for practical purposes.

## Figures and Tables

**Figure 1 biomolecules-09-00408-f001:**
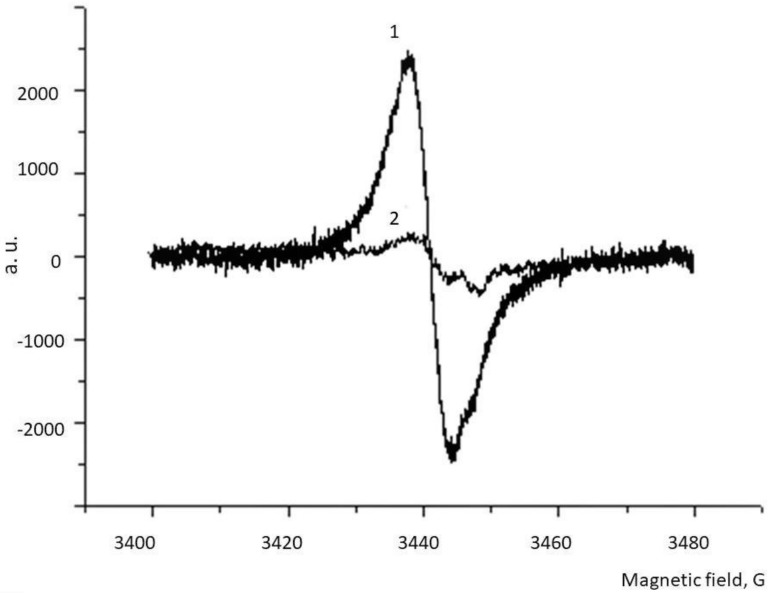
The ESR spectrum of water-soluble melanin from BSF. Curve 1—the initial spectrum; curve 2—after oxidation with 1% hydrogen peroxide solution for 2 h. a. u.—arbitrary units.

**Figure 2 biomolecules-09-00408-f002:**
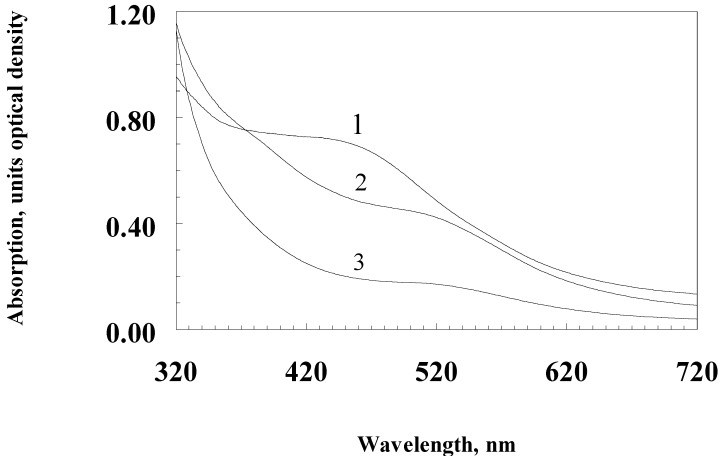
The absorption spectrum of purified *Hermetia illucens* ommochromes in a mixture of 0.1 M potassium phosphate buffer (pH 7.4) and methanol-HCL in a volume/volume ratio of 2:1. Curve 1—initial spectrum; curves 2 and 3—in the presence of 1.5% hydrogen peroxide solution (curve 2—20 min of reaction; curve 3—2 h of reaction).

**Figure 3 biomolecules-09-00408-f003:**
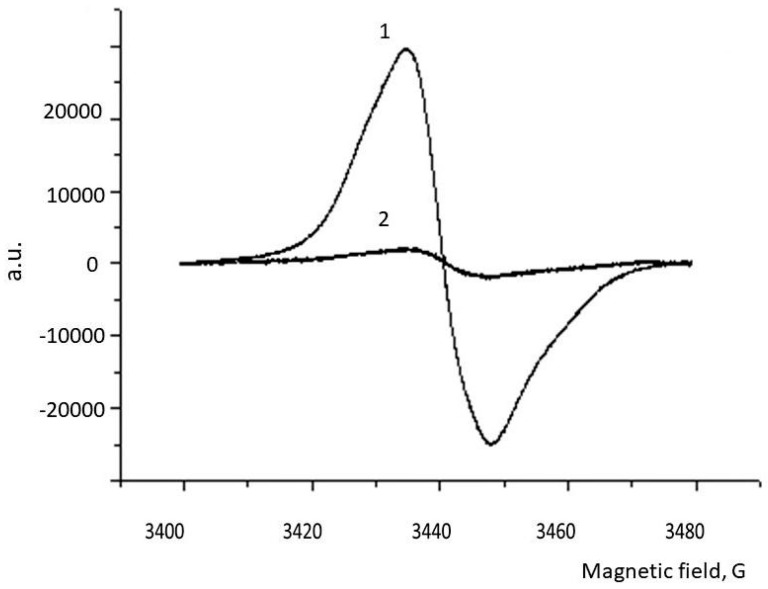
ESR spectrum of ommochromes from BSF. Curve 1—the initial spectrum; curve 2—after oxidation with 1% hydrogen peroxide solution during 2 h.

**Figure 4 biomolecules-09-00408-f004:**
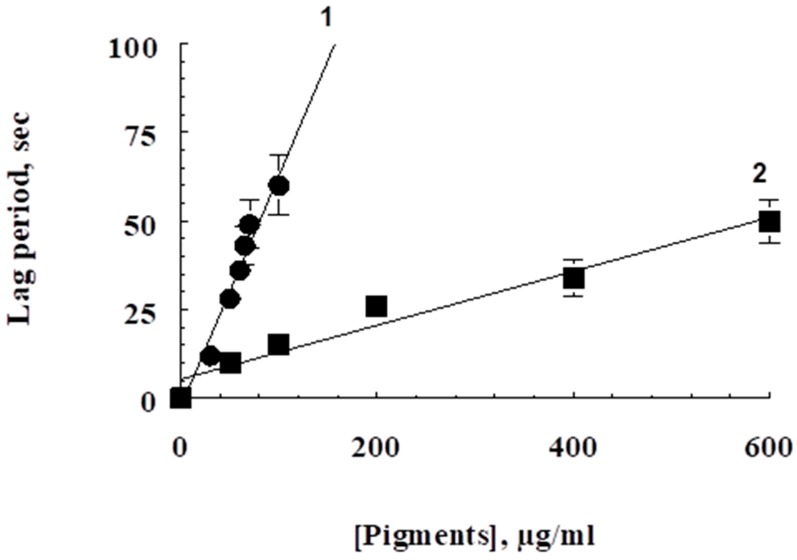
Dependence of the latent period of luminol chemiluminescence on pigment concentration. Curve 1—BSF ommochromes are added; curve 2—BSF water-soluble melanin is added. Data are means (means ± SD) of three assays.

**Figure 5 biomolecules-09-00408-f005:**
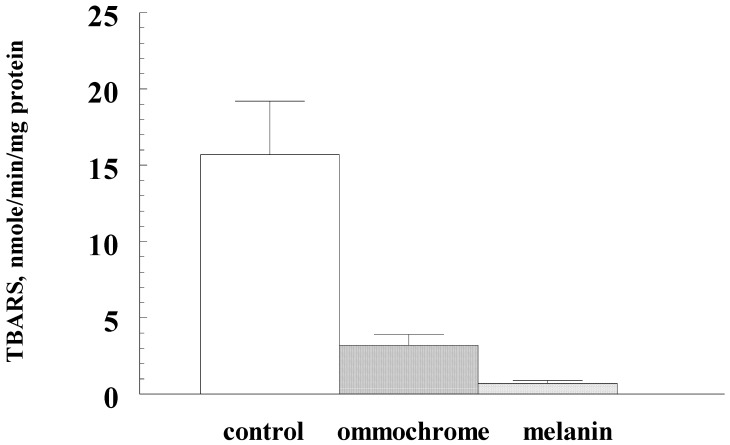
The inhibitory effect of delipidated form of water-soluble melanin (DL-WSM) and BSF’s ommochromes on the peroxidation process of the outer segments of the photoreceptor cells of a bull’s eye. The control—without the addition of pigments; experiment—added 0.35 mg/mL ommochromes and 0.4 mg/mL melanin. The ordinate axis—rate of accumulation of TBA-reactive products (TBARS) in nmol/mg protein per 1 min. Data are means (means ± SD) of three assays, *p* < 0.05.

**Table 1 biomolecules-09-00408-t001:** Electron spin resonance (ESR) characteristics of the dry samples of black soldier fly (BSF) melanin at room temperature.

Melanin Sample	The Concentration of Paramagnetic Centers, 10^17^ spin/g Dry Weight	g-Factor	∆H_pp_
WSM	3.4 ± 0.6	2.0038	6.2
DL-WSM	4.4 ± 0.7	2.0040	6.0
AHM	15.0 ± 1.7	2.0037	5.5
DL-AHM	26.0 ± 2.4	2.0037	5.5
DOPA-melanin	51.0 ± 3.5	2.0027	4.8

Data are means (means ± SD) of four assays. Abbreviation: WSM—water soluble melanin; AHM—chitin-melanin sample; DL-WSM—delipidated form of WSM; DL-AHM—delipidated form of AHM.

**Table 2 biomolecules-09-00408-t002:** Antiradical and sorption activity of the main samples of BSF melanins.

Melanin Samples	Antiradical Activity, μM Ascorbate	Maximum Binding of Methylene Blue (MB), mg MB/g Dry Weight
WSM	180 ± 16	-
DL-WSM	250 ± 19	390 ± 43
AHM	18.0 ± 3.5	-
DL-AHM	32.0 ± 5.0	180 ± 17
DOPA-melanin	6000 ± 460	360 ± 45

Data are means (means ± SD) of four assays. Abbreviation: WSM—water soluble melanin; AHM—chitin-melanin sample; DL-WSM—delipidated form of WSM; DL-AHM—delipidated form of AHM.

**Table 3 biomolecules-09-00408-t003:** ESR characteristics of BSF suspension of ommochromes in 0.1 M K-phosphate buffer (pH 7.4) at liquid nitrogen temperature (measurements were made at 77 K, P = 0.02 mW).

Sample	The Concentration of Paramagnetic Centers, 10^17^ Spin/g Dry Weight	g-Factor	∆H_pp_
Suspension of ommochromes in 0.1 M K-phosphate buffer	16.0 ± 2.4	2.0045	13.2

Data are means (means ± SD) of four assays.

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
