# Peer review of "Antioxidative Properties of Melanins and Ommochromes from Black Soldier Fly Hermetia illucens"

_biomolecules, 2019, doi:10.3390/biom9090408_

Round 1

Reviewer 1 Report

Dear authors,

thank you for your scientific contribution to reveal the function of melanins and ommochromes in BSFs.

Yet, I have quite a few remarks which you might want to consider, especially regarding the first part of your materials & methods section.

l 5: "Severtsov Institute of problems ecology..." - Is this correct?

Abstract

l 13: check space characters

l 14: ommochromes were not mentioned before - you only introduced your study with "a comparative study of melanin-containing samples isolated from..."

l 18: ... was carried OUT

l 20: Hermetia illucens always italic

Introduction

Your introduction starts with general aspects about the importance of biologically active substances from insects. Then you go much into detail and explain Hermetia illucens, afterwards returning to general remarks about melanins and ommochromes. I recommend that you reconsider the structure of the first part of your introduction.

l 28: What do you mean by "national economy"?

l 34: "in general"

l 38 Ommochromes are not absorbed BY ultraviolet light, but they absorb it.

Your statements and corresponding literature reference is not always clear. Please check.

Results

l 53/54: Your WSM and DL-WSM were well solubilized in buffer, you state. --> l 129: I can read about the "poor solubility". So which one is true now?

Table 1: sample size?

Table 2: sample size?

Fig. 1: y-axis label on top, Fig. 2: axis label on left side of y-axis

Fig. 1: WSM of which organism?; Is the oxidation method being described in the M&M part?

l 74: "absorption maximum... approx. under 470 nm" --> Can you give an exact wave length?

l 77/78: Please rephrase your english.

l 79: space character

l 81: "fractions of ommochromes fly" - What do you mean by that?

l 83: "... WHICH was obtained..."

Fig. !/3: Why don´t you combine the figures as they look prettey much alike?

l 84/85: In the figures, no "spectral lines" are shown but first derivatives of the absorption curves.

Fig. 2: "along the ordinate is..." - Please rephrase. (E.e. "is plotted"); change "3-2 hours" to "2-3 hours"

Fig. 3: y-axis label on top; "ESR spectrum of the eye of a BSF´s eye"? - What do you mean by that? Please check.

l 101: Delete "in".

l 104: Substitute "are" by "is".

l 107: Delete "in".

l. 108: Delete "the".

Fig. 4: "Ommochromes of a fly are added. - Please specify, also for curve 2. How many repetitions did you do?

Fig. 5: How many repetitions?, Please check y-axis labelling. DL-WSM of which organism?

Discussion

l 125 - l 129: Contradiction. - Good or poor solubility?

l 130: Reference? Could preparation be optimized?

l 133: What do you mean by "most having prospects"?

l  134: What do you mean by "used... for the introduction of ra materials and finished products"?

l  136-140: Do yo have literature/references available?

l 141-143: Considering the preparation methods, do you truly believe that BSF can be a source of melanins and ommochromes at industrial scale?

L 143: "National economy"?

Materials & Methods

Please reconsider and rephrase section 4.1. Maybe a general scheme would be helpful to the audience.

Where can I find information about where the BSF came from and/or how they were bred?

l 145: "melanin BSF"?

l. 146: "The objects of the study were adult flies of H. il. after breeding under laboratory conditions."

l 147: How old were the flies? Were they alive or did you collect dead ones?

l. 148: Freezing. - How?

l. 149: Mixer. - Model?

l. 150: Ratio for what?; during --> for

l 151: Filtration. - Which filter paper or which device did you use?

l 151/152: Washing with distilled water. --> Did you discard or use it for next step, the precipitation?

l 152: For how long did you dry at 70 °C?

l 156: Which "homogenized biomass"?

l 157: Where do the 20 g of biomass come from?

l 157: "For this... was poured...?" Please rephrase.

l 160: What is a "slight specific odor"?

l 160/l 161: Repetition of "was obtained".

l 161: "which matches..." Please rephrase.

l 161: Yield from what?

l 163: "fat from the obtained preparations" - Which preparations?

l 163: To do this.

l 164: Which one is your "initial sample"?

l 165/166: Repetition of "was removed".

l 166/167: "boiling water bath (60-700 °C)? --> I don´t think you actually prepared your samples like this.

l 174: Flies freshly decapitated? How old were they?

l 174: Absolute MeOH-HCl solution. Please explain.

l 176: Which filter paper?

l 176: There is a verb missing.

l 178: 5000xg

l 178: Which MeOH-HCl solution do you mean here?

l 179: Repeated from where?

l 180: "was washed..."

l 181: Please rephrase "It was received..."

l 181: What do you mean by "true solution"?

l 199: K-phosphate buffer of which concentration?

l 200: Which MeOH-HCl solution?

Author Response

Thank you for fast consideration of our manuscript (biomolecules-546808). I studied your remarks. It seems to me remarks quite reasonable. I would like to thank you for careful studying of our manuscript and for major critical remarks. It always is useful for improvement quality of work. We made all (hopefully) the necessary changes to the text of the manuscript.

Now for the comments.

Line 74. The absorption spectrum of ommochromes, extracted from BSF heads and purified by double reprecipitation, has a maximum (shoulder) in the region of 460-480 nm. On average, it is 470 nm. The wavelength of the maximum (shoulder) varies somewhat depending on the type of solvent (MeOH-HCl or buffer solution).

Sincerely yours,

Dr. Nina Ushakova

Reviewer 2 Report

I highly recommend that the English be improved in this manuscript to facilitate its comprehension. Here is what I noticed:

Line 25: drugs AND pharmaceutical; Also instead of Currently

Line 34: in general

Line 36: of the properties of the pigment and its isolation methods

Line 39: Ommochromes are strongly absorbed by ultraviolet radiation, have characteristic...

Line 44: shielding instead of screening

Line 46: this sentence is out of place.

Line 48-49: isolated is repeated twice.

Line 54: close to

Line 77: please correct "as like as in"

Line 83: the concentration of the melanin-chitosan complex THAT

Line 101 and 107: Fig.  X shows

Line 123: fix this sentence

Line 128: "is not relatively high" is confusing

Line 130: may be associated with the presence of a large amount..

Line 133: bad English.

Line 134: this sentence is out of place

Line 136-143: this whole paragraph looks like a conclusion and not part of a discussion section. Please edit.

Line 181: what do you mean by true solutions?

In addition to this, I need to point out the following issues:

1) The introduction completely lacks any reference to the ESR technique which is heavily utilized in this paper. I would recommend include references and briefly describe why ESR can be successfully used to characterize these classes of biomolecules, considering that some researchers might not be familiar with this technique.

2) Referring to line 54, it is not proven that the suspension is close to molecular dissolution. DLS would be appropriate. If this is not possible, please rephrase.

3) Why does chloroform increase the concentration of paramagnetic centers (line 58)? There is no mention of the molecular basis of it in the discussion.

4) What is the conclusion at the molecular level drawn by observing the H2O2-induced oxidation of melanin? Why was this experiment carried out to strengthen the conclusions of the paper? Please explain.

5) Lines 75-79 contain too many assumptions that are not supported by either references or experimental evidence. I recommend include appropriate references to your conclusions such as, Panettieri et al, PLOS One 2018; Stavenga et al, Journal of Experimental Biology 2014; Riou et al, Journal of Chemical Ecology 2010.

Also, referring to the UV-Vis spectrum of the ommochrome crude extract is not sufficient to unmistakably identify the chemical nature of the main component(s) and state that hydrogen peroxide leads to the formation of xanthommatine (it is indeed very unlikely in such conditions), as the peaks are very broad and the crude extract most likely contains a more or less complex mixture of xanthommatine derivatives (see the recommended references) and other compounds. The UV-Vis spectrum #1 in Fig 2 could easily correspond to several other substituted xanthommatine whose absorption maxima are very close to the ones of xanthommatine itself. Furthermore, the background of these spectra looks quite high and the #1 and 2 show a non-negligible absorption beyond 600 nm which is not usually observed for xanthommatine derivatives. Please provide HPLC-MS analysis or other suitable analytical methods in addition to these UV-Vis spectra to justify your conclusions. Perhaps even more importantly, in the materials and methods section you do not provide a reference for the ommochrome extraction protocol. In absence of it, a more rigorous characterization of the crude extract is absolutely necessary.

6) Related to the previous comment, the melanin extraction protocol does not have any reference which makes the soundness of the overall results quite shaky. Also, from my perspective the harshness of the protocol using sulfuric acid makes me doubt the purity of the melanin extract. For instance, why should a melanin extract of a acceptable level of purity look like a oily powder with a specific odor? In absence of any in-depth characterization of the crude extract, it is necessary to provide at least a reference supporting this protocol. 

7) Fig 4 lacks repeats and corresponding error bars. Please provide.

8) The experimental protocol related to Fig 5 lacks the number of repeats that were used to calculate the error bars and the kind of statistical analysis that was employed. Please provide this information in the materials and methods section. 

9) The discussion needs to address all the findings expounded in the results section. The current version makes the main objective of this paper quite confusing to the reader. Please edit and re-organize more clearly.

10) Since there is no trace of the molecular structures of the compounds studied in this work, I would suggest to change the title of the paper. As is, it is too broad and deceiving as one would expect to find the molecular structures of melanins and ommochromes of Black Soldier Fly. Please suggest an alternative title.

Author Response

Thank you for fast consideration of our manuscript (biomolecules-546808). I studied your remarks. It seems to me remarks quite reasonable. I would like to thank you for careful studying of our manuscript and for major critical remarks. It always is useful for improvement quality of work. We made all (hopefully) the necessary changes to the text of the manuscript.

I agree with you that the title of our study is too broad. Since our work is mainly devoted to the research of the antioxidant properties of the black soldiers fly Hermetia illucens pigments - melanins and ommochromes, we propose a new title of study: “Antioxidative properties of melanins and ommochromes from black soldiers fly Hermetia illucens”.  We made changes (edits) to the introduction, adding new references, to the methodological part (to clarify the protocol for obtaining (isolation) melanins and ommochromes, as well as to the “Results” chapter.

I agree with your recommendation that the manuscript requires an improvement in English.

Now for some comments.

Line 44. Ommochromes in the literature are commonly referred to as screening pigments and not as shielding pigments.

Point 3 (in addition).  I think that chloroform increases the relative concentration of melanin in the sample DL-AHM, since it removes impurity lipids and the EPR signal increases due to a decrease in the denominator (i.e., the dry weight of the sample). I consider it inappropriate to include this in the discussion.

Point 4 (in addition).  It is known that melanins and ommochromes are destroyed in the presence of hydrogen peroxide, thus losing the EPR signal. In my opinion, this fact also characterizes our samples. Necessary references are included in the manuscript.

Point 5 (in addition) The methanol-HCl extract obtained from the “black soldiers” fly heads undoubtedly contain ommochromes, but not any other pigments. This is evidenced not only by their solubility in acidic methanol, but also by many of their other properties, namely:

a) these pigments are not extracted by neutral alcohols, alkalis or organic solvents, but are extracted only with acidic alcohols and formic acid; b) pigments are not soluble in common organic solvents, poorly soluble in water, highly soluble in acidic alcohols and formic acid; c) pigments are stable in acidic solutions and unstable in alkaline solutions; d) pigments are destroyed by high concentrations (5–10%) of hydrogen peroxide.

All these characteristics (properties) of ommochromes were established in the works of Becker (1939, 1942) and followers. See, for example, B. Ephrussi and J.L. Herold “Studies of eye pigments of Drosophila. I. Methods of extraction and quantitative estimation of the pigment components”, Genetics, 1944, V. 29, P. 148-175.

We have repeatedly used these methods for isolating and identifying ommochromes from the eyes of various invertebrates, mainly crustaceans, to study their physicochemical characteristics and antioxidant activity. See for example: Dontsov, A. E. (1981) Antioxidative function of shrimp ommochromes Pandalus latirostris.  J. Evolut. Biochem. Physiol., v. 17, pp. 53–56 (in Russian); Dontsov, A. E.; Mordvintsev, P. I.; Lapina, V. A. (1985) Dark and light induced ESR signals of the ommochromes of the invertebrate eye. Biophysics, v. 30(1), pp, 6–8 (in Russian); Sakina, N. L.; Dontsov, A. E.; Lapina, V. A.; Ostrovsky, M. A. (1987) The system of protection of the eye structures from photodamage. Screening pigments of arthropods – ommochromes as inhibitors of photo-oxidation processes. J. of Evolutionary Biochem. Physiol., v. 23(6), pp. 702–706 (in Russian); Ostrovsky, M. A., Sakina N.L., Dontsov A.E. (1987) Antioxidative role of eye screening pigments. Vision Res., v. 27(6), pp. 893-899; Pustynnikov, M. G.; Dontsov, A. E. (1988) Inhibition of UV-induced accumulation of lipid peroxides by melanins and ommochromes.  Biochemistry, v. 53(7), pp. 1117–1120 (in Russian); Dontsov, A. E.; Ostrovsky, M. A. (2005) The antioxidant role of shielding eye pigments – melanins and ommochromes, and physicochemical mechanism of their action. Chemical and Biological Kinetics. New Horizons. [Eds.: S. D. Varfolomeev, & E. B. Burlakova,]. Danvers; MA 0193, USA, v. 2, pp. 133–150; Dontsov, A. E.; Fedorovich, I. B.; Lindström, M.; Ostrovsky, M. A. (1999). Comparative study of spectral and antioxidant properties of pigments from the eyes of two Mysis relicta (Crustacea, Mysidacea) populations, with the different light damage resistance. J. Comp. Physiol. B., v. 169, pp. 157–164.

Of course, the extract obtained by us from the “black soldiers” fly contains a mixture of ommochromes, as can be seen from the broadband absorption spectrum. In this case, the pigments dissolved in neutral methanol were removed by us. Now the study on determining the qualitative composition of ommochromes in this extract from the black soldiers fly by HPLC is carried out in our laboratories. We would be happy if some previously unknown substance of red color was actually present in this extract. If we find, be sure to write about it in your journal. But in this study the main result for us was that we obtained in large quantities a dry preparation containing ommochromes and exhibiting high antioxidant activity from the purified extract of the “black soldier” fly. Our assumption that xanthommatin is the main component in this mixture of ommochromes is based on the fact that the eyes of flies possess usually only xanthommatin. So, for a fly Calliphora erythrocephala is shown that only xanthommatin is contained in their complex eyes (The complete book on Natural Dyes & Pigments, 2005, NIIR Board of Consultants & Engineers, Asia Pacific Business Press Inc. ISBN: 8178330326, p. 448). Whether the other ommochromes contains in a “black soldier” fly besides xanthommatin is the subject of our future research.

Sincerely yours,

Dr. Nina Ushakova

Reviewer 3 Report

In this study, the authors compare the antiradical and antioxidant properties of pigments from the black soldier fly (BSF). They study melanins and ommochromes in different chemical conditions, and found that uncomplexed delipidated melanins are the best source of antioxidants. They conclude that BSF, as a potential food and feed insect resource, could provide natural antioxidants.

Broad comments

This study is interesting by the experimental methods that are employed to answer a long standing question, are all pigments potent antioxidants? To that purpose, the authors characterized the ESR, sorption, antiradical and biological antioxidant properties of extracted pigments from BSF. However, the way the authors analyzed and concluded on their data suffers from major methodological flaws. 

First, and most importantly, the authors did not provide the number of biological replicates (n = ?), they did not explain the type of variation they calculated (standard deviation, standard errors ?) and they did not perform any statistical tests to support their conclusions. I would have expected, at least, ANOVAs on the data presented in Tables 1, 2 and in Fig. 5.

Second, although the authors described quite precisely their protocols of pigment extraction, it is unclear why they performed all those steps and what is the rational behind each one of them. I am not aware of any study describing the presence of free lipidated melanins and chitin-complexed lipidated melanins in BSF. The authors should either provide evidence to their claim or give references to previous studies (on this system or at least in any other insect). I would do the same comment for the ommochromes. The only evidence that the pigments they extracted belong to ommochromes is that they came from insect eyes and that they are soluble in acidified methanol (which is particularly true for ommatins, it is far less clear for ommins). The absorbance spectra provided is of no use to determine the class of extracted pigments (see the specific comments below).

Lastly, the authors do not relate their results with studies from other groups sufficiently. In particular, during the last five years, theoretical results on the antiradical of ommochromes have been obtained (see for exemple "Romero, Y., & Martínez, A. (2015). Antiradical capacity of ommochromes. Journal of molecular modeling, 21(8), 220." and "Zhuravlev, A. V., Zakharov, G. A., Shchegolev, B. F., & Savvateeva-Popova, E. V. (2016). Antioxidant properties of kynurenines: density functional theory calculations. PLoS computational biology, 12(11), e1005213."). These studies are of particular importance here, they should be cited and discussed. The same may be true for melanins, I am quite sure other groups have worked on the theoretical or experimental antiradical properties of these pigments.

Furthermore, the manuscript is hard to follow because of a lack of methodological explanation. ESR experiments should be explained and the type of measurement the authors made (concentration of paramagnetic centers, g-factor and ΔHpp) should be related to antiradical properties. Otherwise, these data would not be usable nor understandable by most readers. Also, for the sake of clarity, the manuscript would gain from being read and corrected by a native English speaker.

Specific comments

Title: It should be more informative. I think the fact that the authors studied the antiradical properties of two types of pigments is an important information that should appear in the title.

Abstract: it lacks one or two sentences of contextualization. What is the question addressed by the authors, what is the purpose of this study and in what context?

L32: I strongly discourage citing the review from Shamim et al. (2014). It is full of mistakes and errors, such as the names of ommochromes and their biosynthetic pathways. Although it is dated, the best current reference, to my knowledge, is the book of Needham "The significance of Zoochromes" (1974).

L38: misspelling "kinurenin" --> "kynurenine". Furthermore, the authors wanted to say that "ommochromes absorb strongly UV", not the other way around.

L39: the absorption band between 430-530 nm is not so characteristic of ommochromes. Their absorption maxium is indeed in this range depending on their redox states but the same absorption band would exist for any bright yellow, red or purple pigment. This sentence should be rephrased in accordance. Adding references to recent reviews or article on ommochromes would also help readers that are not familiar with this class of pigments. In general, the introduction would gain from having more references.

L41: misspelling "ommatines" --> "ommatins" and "ommines" --> "ommins". I do not agree with both ommatins and ommins appearing dark in vivo. This is true for any pigment as long as it is sufficiently concentrated. But in most cases, when ommins are not present, the colors of ommatin-containing tissues are either yellow, orange or red. But not to the point of being black.

L47: the authors should cite the studies of Romero and Martinez 2015 and Zhuravlev et al. 2016 that investigated the theoretical antiradical properties of ommochromes. They are a good starting point for the introduction on the role of pigments as natural antioxidants.

L52: we need references or results that support the claim that melanin samples contain chitin and lipids.

L54: could the authors provide evidence for their claim of having a suspension closing to molecular solution? Such as photographs?

L55: ESR needs to be defined. The technique, the measurements and the results obtained should also be explained and put in te context of the study. Why are ESR measurements relevant to the study of antiradical capacity? This is a question any person unfamiliar with this technique would ask.

L64: the authors need to explain why they performed sorption activity experiments. What do they tell us about the antiradical capacity of pigments, especially when it is done with methylene blue?

Table 1: all acronyms should be defined for the sake of clarity. DOPA should be referred to as DOPA-melanin since it is not pure DOPA that has been tested bu synthetic melanin from DOPA. The authors should define the type of variation they calculated (+/-SE? +/-SD?). The number of replicates should also be indicated. Statistical tests, such as an ANOVA, should be applied on this set of data to support the conclusions.

Table 2: idem as Table 1.

Figure 1: all acronyms should be defined for the sake of clarity.

L73-75: I am not convinced at all by the conclusion that methanolic extracts of BSF contain ommochromes, even less dihydroxyxanthommatin. The only evidence provided by the authors is an absorbance spectrum of the raw extraction showing an increasing absorption from high wavelength to low wavelength, with a sort of "bump" around 470 nm. This is definitely not the characteristic absorption spectrum of dihydroxanthommatin. We only see here the absorption band typical of a red pigment. With the current evidence, it is impossible to link it to an ommochrome (unless the authors have references of studies that performed such extractions on BSF eyes and successfully identified dihydroxanthommatin by TLC, HPLC-UV or better MS/MS). The authors need to address this point before being able to say that their pigment extracts contain ommochromes. I would recommend injecting the extract in a HPLC with a PDA detector, for example. At least, TLC with ommatins synthesized according to the procedure of "Butenandt, A., Schiedt, U., & Biekert, E. (1954). Über Ommochrome, III. Mitteilung:. Synthese des Xanthommatins. Justus Liebigs Annalen der Chemie, 588(2), 106-116." should be feasible if no HPLC system is available.

L75-80: misspelling, there is no final "e" at "dihydroxanthommatin", "xanthommatin" and "ommatin"

L76-79: The effect of H2O2 on the absorbance spectrum reminds me more of a bleaching effect than a simple oxidation of the red dihydroxanthommatin into the yellow xanthommatin. If it was the case, the authors would not have obtained a global and sharp decrease of absorption, only a decrease around 470 nm and probably an increase around 440 nm, which is the absorption maximum of xanthommatin. Again, the authors need to provide more analytical evidence to support their conclusion on the type and family of pigments.

Figure 3: same as Figure 1

Table 3: same as Table 1. Furthermore, the authors do not explain why they did not perform antiradical and sorption activity experiments on ommochromes, while their goal was to compare both ommochromes with melanins. Why are the conditions of ESR measurements provided in the title of this table and not for melanins? Are the conditions different and, if so, could it impact the comparison?

L107-109: in the absence of information about the number of replicates and of statistical tests, the authors cannot simply reach the conclusion that pigments are effective biological antioxidants and that melanins are better than ommochromes.

L109-110: could the authors provide references to their claim that the high sorpion acitivity of melanins (although they did not test it for ommochromes) and the binding of metals could explain the better antioxidant property of melanin? And also discuss it in more details in the Discussion section.

Figure 4: could the authors tell us how they performed the linear regression? Are there statistical results related that would help comparing the two curves?

Figure 5: I do not understand the vertical axis, what is the unit here? nmol/min/mg? Also, TBARS should be defined somewhere in the legend. As already mentioned, this figures lacks the number of replicates, the type of error bars and statistical tests. Furthermore, it is better to plot each individual point along with the histogram, so the reader can fully appreciate the variability of results. Could the authors tell why they used 0.35 mg/mL of ommochromes vs. 0.4 mg of melanins? Why a difference of 0.05 mg between the two pigments? Could it explain the results they obtained, meaning that melanins are better antioxidants compared to ommochromes only because they were more concentrated?

Discussion: this section contains only one reference, which is surely not enough for discussing the importance of the results obtained. The authors should better take into account the work from other groups on the antioxidant of melanins, ommochromes and probably other pigments. I already pointed out some references, but other may also exist and prove useful to the discussion section of this manuscript. Another example is found L137-138, the authors do not explain why they consider ommochromes as potent antimocribial and antitumor agents. Overall, it would help the reader to understand how important it is to have experimental data on the antiradical capacities of pigments from food and feed insects.

L150: it is unclear what the ratio 1:4 refers to. Four times more of homogenate or of 10% NaOH? Relative to their weight or volume?

L152: the authors should refer to a study that demonstrates this protocol leads to a chitin-melanin complex

L155: idem for the water-soluble melanin

L167: misspelling I guess "60-700 °C" --> "60-70 °C"

Author Response

Thank you for fast consideration of our manuscript (biomolecules-546808). I studied your remarks. It seems to me remarks quite reasonable. I would like to thank you for careful studying of our manuscript and for major critical remarks. It always is useful for improvement quality of work. We made all (hopefully) the necessary changes to the text of the manuscript.

I agree with you that the title of our study is too broad. Since our work is mainly devoted to the research of the antioxidant properties of the black soldiers fly Hermetia illucens pigments - melanins and ommochromes, we propose a new title of study: “Antioxidative properties of melanins and ommochromes from black soldiers fly Hermetia illucens”.  We made changes (edits) to the introduction, adding new references, to the methodological part (to clarify the protocol for obtaining (isolation) melanins and ommochromes, as well as to the “Results” chapter.

I agree with your recommendation that the manuscript requires an improvement in English.

Now for some comments.

The methanol-HCl extract obtained from the “black soldiers” fly heads undoubtedly contain ommochromes, but not any other pigments. This is evidenced not only by their solubility in acidic methanol, but also by many of their other properties, namely: a) these pigments are not extracted by neutral alcohols, alkalis or organic solvents, but are extracted only with acidic alcohols and formic acid; b) pigments are not soluble in common organic solvents, poorly soluble in water, highly soluble in acidic alcohols and formic acid; c) pigments are stable in acidic solutions and unstable in alkaline solutions; d) pigments are destroyed by high concentrations (5–10%) of hydrogen peroxide.

All these characteristics (properties) of ommochromes were established in the works of Becker (1939, 1942) and followers. See, for example, B. Ephrussi and J.L. Herold “Studies of eye pigments of Drosophila. I. Methods of extraction and quantitative estimation of the pigment components”, Genetics, 1944, V. 29, P. 148-175.

We have repeatedly used these methods for isolating and identifying ommochromes from the eyes of various invertebrates, mainly crustaceans, to study their physicochemical characteristics and antioxidant activity. See for example: Dontsov, A. E. (1981) Antioxidative function of shrimp ommochromes Pandalus latirostris.  J. Evolut. Biochem. Physiol., v. 17, pp. 53–56 (in Russian); Dontsov, A. E.; Mordvintsev, P. I.; Lapina, V. A. (1985) Dark and light induced ESR signals of the ommochromes of the invertebrate eye. Biophysics, v. 30(1), pp, 6–8 (in Russian); Sakina, N. L.; Dontsov, A. E.; Lapina, V. A.; Ostrovsky, M. A. (1987) The system of protection of the eye structures from photodamage. Screening pigments of arthropods – ommochromes as inhibitors of photo-oxidation processes. J. of Evolutionary Biochem. Physiol., v. 23(6), pp. 702–706 (in Russian); Ostrovsky, M. A., Sakina N.L., Dontsov A.E. (1987) Antioxidative role of eye screening pigments. Vision Res., v. 27(6), pp. 893-899; Pustynnikov, M. G.; Dontsov, A. E. (1988) Inhibition of UV-induced accumulation of lipid peroxides by melanins and ommochromes.  Biochemistry, v. 53(7), pp. 1117–1120 (in Russian); Dontsov, A. E.; Ostrovsky, M. A. (2005) The antioxidant role of shielding eye pigments – melanins and ommochromes, and physicochemical mechanism of their action. Chemical and Biological Kinetics. New Horizons. [Eds.: S. D. Varfolomeev, & E. B. Burlakova,]. Danvers; MA 0193, USA, v. 2, pp. 133–150; Dontsov, A. E.; Fedorovich, I. B.; Lindström, M.; Ostrovsky, M. A. (1999). Comparative study of spectral and antioxidant properties of pigments from the eyes of two Mysis relicta (Crustacea, Mysidacea) populations, with the different light damage resistance. J. Comp. Physiol. B., v. 169, pp. 157–164.

Of course, the extract obtained by us from the “black soldiers” fly contains a mixture of ommochromes, as can be seen from the broadband absorption spectrum. In this case, the pigments dissolved in neutral methanol were removed by us. Now the study on determining the qualitative composition of ommochromes in this extract from the black soldiers fly by HPLC is carried out in our laboratories. We would be happy if some previously unknown substance of red color was actually present in this extract. If we find, be sure to write about it in your journal. But in this study the main result for us was that we obtained in large quantities a dry preparation containing ommochromes and exhibiting high antioxidant activity from the purified extract of the “black soldier” fly. Our assumption that xanthommatin is the main component in this mixture of ommochromes is based on the fact that the eyes of flies possess usually only xanthommatin. So, for a fly Calliphora erythrocephala is shown that only xanthommatin is contained in their complex eyes (The complete book on Natural Dyes & Pigments, 2005, NIIR Board of Consultants & Engineers, Asia Pacific Business Press Inc. ISBN: 8178330326, p. 448). Whether the other ommochromes contains in a “black soldier” fly besides xanthommatin is the subject of our future research. The use of pigments with slightly different concentrations (0.35 mg/ml of ommochromes and 0.40 mg/ml of melanin) in these experiments (Fig.5) was technically random. However, this difference in concentration practically has no effect on the final result. An increase in the concentration of melanin relative to the concentration of ommochromes, which is only 13%, could lead to an increase in its inhibitory effect, also in approximately the same proportion, since inhibition linearly depends on the concentration (Fig. 4, curve 2 for the latent period of quenching of chemiluminescence). In the experiment (Fig. 5) melanin is almost three times better than ommochromes slows down the rate of accumulation of TBA-reactive products. In this paper, we presented data on the sorption activity of melanin BSF in order to show that these pigments can be used for practical purposes and as an effective biosorbents. Since ommochromes are not polymeric substances, such experiments were not performed for them.

Sincerely yours,

Dr. Nina Ushakova

Round 2

Reviewer 2 Report

Dear authors,

Thank you for editing the manuscript according to some of my suggestions. The paper has definitely improved.

I would still like to see more recent references among the ones you mentioned in your response which are all quite old, considering that there have been a plethora of papers about this topic. The three articles I mentioned in my previous review would certainly be a good addition. Of particular interest is the Panettieri et al. paper in Plos ONE that shows the presence of several xanthommatin derivatives (though in a different organism) revealed through HPLC-MS analysis. This paper might also be of interest to your future studies.

Please include some information about ESR in the introduction as I mentioned previously. I saw you added a sentence in the EPR paragraph at the end of the paper but I do not think it is enough. Readers should have a more clear idea about the technique since the beginning, otherwise the significance of the experiments is not of easy comprehension.

Also, the discussion looks better now but it is still a bit disconnected. Try to mention the use of your protocol for the production of antioxidants only at the very end of the discussion. Before that, focus on discussing the data more rigorously.

I look forward to your newly edited version of the manuscript.

Thank you

Reviewer 3 Report

I wish to thank the authors for the time they took to answer my criticisms and comments. The authors have dealt with some of my major and minor concerns, although I still have comments to improve the manuscript even further.

1. I appreciate that the authors indicated sample sizes and standard deviation for each experiment, but writing in the Materials and Methods that Student's t-tests were applied is not sufficient. In which experiment were these tests applied? What were the results and p-values associated? Depending on the experiment, would an ANOVA (as already stated in my previous report) be more suitable?

2. I agree with the authors that their acidified methanolic extracts most likely contain ommochromes, and I appreciate the fact that they discuss in length why there is possibly dihydroxanthommatin, although other ommochromes could be involved. However, I would suggest not to show the raw absorbance spectrum of ommochromes, but the absorbance spectrum after substracting the absorbance spectrum background (solution/buffer + cuvette). I think this would enhance the resolution of the peaks and shoulders in the visible region. It would hence resemble more closely to an ommochrome spectrum. This experiment (which should have been done and described more precisely in the Materials and Methods anyway) does not require a lot of time.

3. I regret that the ESR parameters given in tables were not explained more in details as I requested. I still think it would greatly improve the readability of the manuscript.

I also have some minor comments and corrections:

L39: misspelling "kinurenine" --> "kynurenine"

L40: misspelling "Omochromes" --> "Ommochromes", "ultraviole" --> "ultraviolet"

L43: misspelling "Omochromes" --> "Ommochromes"

L48: misspelling "kinurenin" --> "kynurenine"

Figure 5: the legend for the Y-axis has not been corrected, it is still misleading whether we should read nmol/min/mg or any other unit.

L156: ommochromes and melanins are not organelles, but pigments. The authors should rephrase this sentence.
